# Prevalence and antibiotic resistance of *Escherichia coli* in urban and peri-urban garden ecosystems in Bangladesh

Pritom Kumar Pramanik[1], M. Nazmul Hoque[2], Md. Liton Rana[1], Md. Saiful Islam[1], Md. Ashek Ullah[1], Fahim Haque Neloy[1], Srinivasan Ramasamy[3], Pepijn Schreinemachers[4], Ricardo Oliva[3], Md. Tanvir Rahman[1]*

1 Faculty of Veterinary Science, Department of Microbiology and Hygiene, Bangladesh Agricultural University, Mymensingh, Bangladesh, 2 Department of Gynecology, Molecular Biology and Bioinformatics Laboratory, Obstetrics and Reproductive Health, Bangabandhu Sheikh Mujibur Rahman Agricultural University, Gazipur, Bangladesh, 3 World Vegetable Center, Shanhua, Taiwan, 4 World Vegetable Center, Bangkok, Thailand

☯ These authors contributed equally to this work.
* tanvirahman@bau.edu.bd

**Data Availability Statement:** All relevant data are within the paper and its Supporting Information files. The minimal dataset necessary to replicate our study findings is publicly available in

## Abstract

In the past decade, there has been a notable rise in foodborne outbreaks, prominently featuring *Escherichia coli* as a primary pathogen. This bacterium, known for its prevalence in foodborne illnesses and as a reservoir of antimicrobial resistance, was isolated from raw vegetables, soil, and water samples collected from rooftop and surface gardens in urban (Dhaka North City Corporation; DNCC and Dhaka South City Corporation; DSCC) and peri-urban (Gazipur City Corporation; GCC) areas of Bangladesh. In this study, 145 samples including vegetables (n = 88), water (n = 27) and soils (n = 30) from DNCC (n = 85), DSCC (n = 30), and GCC (n = 30) were analyzed to assess the prevalence of *E. coli* using culture, biochemical tests, and PCR targeting the *malB* gene. *E. coli* was detected in 85 samples, indicating an overall prevalence of 58.62% (95% CI: 50.48–66.31). In urban areas (DNCC and DSCC), the prevalence rates were 44.70% and 80.0%, respectively, with surface gardens showing higher contamination rates (70.83%) than rooftop gardens (46.57%). In the peri-urban GCC, overall prevalence of *E. coli* was 76.7%, with rooftop gardens more contaminated (93.33%) than surface gardens (60.0%). Antibiogram profiling of 54 randomly selected isolates revealed 100% resistance to ampicillin, with varying resistance to ciprofloxacin (25.92%), tetracycline (14.81%), cotrimoxazole (14.81%), imipenem (9.25%), and fosfomycin (1.0%). Notably, all isolates were susceptible to ceftazidime, gentamicin, chloramphenicol, nitrofurantoin, and cefotaxime. Multidrug resistance (MDR) was found in 14.81% of isolates. The *blaTEM* gene was present in 81.48% of the isolates, while the *tetA* gene was detected in 3.70%. These findings underscore the urgent global health concern posed by the significant presence of *E. coli* in fresh vegetables, highlighting the need for improved safety measures and monitoring to prevent the spread of antimicrobial resistance through the food chain.

Supporting Information Files. Additional information regarding data and materials can be requested from the corresponding author.

**Funding:** This work was conducted as part of the CGIAR Research Initiative on Resilient Cities Through Sustainable Urban and Peri-urban Agri-food Systems and is supported by contributors to the CGIAR Trust Fund (https://www.cgiar.org/funders).

**Competing interests:** NO authors have competing interests.

## Introduction

Vegetables are regarded as vital components of balanced diets due to the phytochemicals, vitamins, minerals, and dietary fiber they provide. Vegetables in the daily diet have been associated in a significant way with improved gastrointestinal health, enhanced vision, a decreased risk of cardiovascular disease, stroke, chronic diseases including diabetes, and certain types of cancer [1,2]. It is believed that certain phytochemicals found in vegetables reduce the risk of chronic disease by preventing free radical damage, influencing metabolic activation and detoxification of carcinogens or even regulating processes that alter the progress of tumor cells [3]. The various vegetables could provide defense against chronic diseases to human beings [4]. Recent research indicates a negative association between vegetable consumption and mortality rates, particularly in cardiovascular disease and cancer [5,6]. However, findings have varied. Some studies suggest a lower mortality risk with increased vegetable intake, yet a British study found no significant mortality differences between vegetarians and non-vegetarians [7,8]. Unbalanced diets, marked by insufficient intake of complex carbohydrates, dietary fiber, and vegetables, account for approximately 2.7 million deaths each year [9] and are among the top 10 risk factors for mortality [4].

While fresh vegetables provide many health benefits, they can also pose potential risks [10]. In recent years, fresh fruits and vegetables have been associated with various outbreaks of transmissible diseases around the world. Efforts are underway to tackle these food safety challenges [11]. Occasionally, raw salad vegetables are consumed without washing, peeling, or applying any heat treatment. This exposes consumers to potential risks of foodborne illnesses [12]. Vegetable contamination can occur at any stage, both before and after harvesting. Using untreated effluent and manure as fertilizers in vegetable cultivation contributes to this contamination [13,14]. Furthermore, a variety of potential contamination sources exist, including debris, animal and human waste products, and the use of contaminated transportation and handling processes. Harvesting and processing equipment can also introduce contaminants. Each of these stages, from the field to the consumer, poses risks for introducing harmful substances into the food supply [15,16]. Earlier studies reported that consumption of a variety of contaminated vegetables and fruits has been linked to outbreaks of viruses, bacteria, and parasites [16,17]. Fecal microorganisms have the potential to endure prolonged periods in soils and manure and water and therefore, they serve as an accessible source of contamination [18,19]. Antimicrobial resistance has become a noteworthy economic and public health worry [20]. Recently, antibiotic resistance in *E. coli* and *Salmonella* spp. has been reported globally [21,22]. *E. coli* keeps getting progressively more difficult to treat as resistance to most first-line antimicrobials has evolved [22]. Furthermore, resistant *E. coli* tends to transfer genes encoding resistance to antibiotics to other strains of *E. coli* and bacteria residing in the gastrointestinal tract, thereby developing resistance from external organisms [23,24]. Resistance to ampicillin, a semi-synthetic-lactam antibiotic commonly used to treat *E. coli* infections in humans and livestock, has recently increased [25]. The prevalence of multidrug resistance *E. coli* in humans and animals is rising worldwide [20,21,26]. The increasing resistance of *E. coli* to beta-lactam antibiotics is leading to severe troubles among the general population [27]. *E. coli* may also develop resistance to various classes of commonly prescribed antibiotics, including trimethoprim-sulfamethoxazole, aminoglycosides, and fluoroquinolones [28]. Such resistance would result in higher mortality and morbidity rates, prolonged hospital stays, elevated treatment expenditures, and disintegration of healthcare facilities. Two plasmid-mediated beta-lactam enzymes, extended-spectrum beta-lactamases (ESBLs) and AmpC beta-lactamases (AmpC), trigger resistance to beta-lactam antibiotics, resulting in a grave effect on the global health sector [28]. Plasmid-mediated AmpC (CMY-2) is a major threat to public health that appears frequently in *Enterobacteriaceae*, especially *E. coli*, in humans and animals [29].

The *malB* gene is involved in the maltose and maltodextrin transport system in bacteria, particularly in *E. coli* [30]. The *malB* operon encodes components essential for the transport and metabolism of maltose and maltodextrins, which are polysaccharides derived from starch. This operon is part of the ATP-binding cassette (ABC) transporter family and includes genes like *malE*, *malF*, and *malG*, which encode for the maltose-binding protein (MalE) and the membrane components (MalF and MalG) that form the maltose transporter complex [30,31].

The presence of multidrug resistance *E. coli* in vegetables is a serious global concern [19]. Despite extensive research on *E. coli* in commercially sourced vegetables [32,33], there is a significant knowledge gap regarding its presence in vegetables from home gardens, which are cultivated organically without pesticides or herbicides. Currently, there is a lack of data on *E. coli* in rooftop and surface gardening practices in Bangladesh. This study aims to isolate and identify *E. coli* from various vegetables, soil, and water samples from rooftops and surface gardens. We also assessed the resistance profiles of the *E. coli* isolates and identified the genes responsible for beta-lactams and tetracycline resistance.

## 2. Materials and methods

### 2.1 Sample information

This study was carried out at the Bacteriology Laboratory, Department of Microbiology and Hygiene, Bangladesh Agricultural University, Mymensingh, Bangladesh, from September 2022 to March 2023. Vegetables, water, and soil samples (S1 Table) were collected from urban locations including Dhaka North City Corporation (DNCC, 23˚52'55.5"N, 90˚24'14.9"E; total population: 5,979,537, total areas: 19,700 hectares) and Dhaka South City Corporation (DSCC, 23˚43'27.0"N, 90˚28'51.0" E; total population: 4,299,345, total area: 10,920 hectares), as well as the peri-urban area of Gazipur City Corporation (GCC, 25˚35'37.1"N, 83˚34'53.4"E; total population: 1,129,145, total area: 32,923 hectares) within the Dhaka division of Bangladesh (S1 Fig) [34]. These areas are approximately 15–20 km apart from each other and feature tropical wet and dry climates. A total of 145 samples including 85 from DNCC (rooftop garden = 43, surface garden = 42), 30 from DSCC (rooftop garden = 15, surface garden = 15) and 30 from GCC (rooftop garden = 15, surface garden = 15) were collected. Further classification of the samples included most commonly grown 88 vegetables namely Coriander (*Coriandrum sativum*), Red amaranth (*Amaranthus cruentus*), Radish leaves (*Raphanus sativus*), Green chilies (*Capsicum frutescens*), Tomato (*Solanum lycopersicum*), Malabar spinach (*Basella alba*)), 27 water samples (Deep tubewell, stored water), and soils (n = 30).

### 2.2 Isolation and identification of *E. coli*

Isolation and identification of *E. coli* were performed by culturing on Eosin Methylene Blue (EMB) agar plates (HiMedia, India) followed by Gram staining. A single loopful of overnight culture grown in nutrient broth was streaked onto EMB agar (HiMedia, India) and incubated aerobically overnight at 37˚C [35,36]. Phenotypic identification of the isolates (N = 85) was performed based on the colony morphology and Gram-staining (Gram -ve, formation of green metallic sheen on EMB), and biochemical tests such catalase, indole, methyl red, Voges-Proskauer (VP), oxidase, urease and triple sugar iron tests [21]. The isolates were molecularly confirmed as *E. coli* using species-specific polymerase chain reaction (PCR) amplification of the *malB* gene (S2 Fig) [21,37]. The *malB* gene specific primers are presented in Table 1. Genomic DNA from overnight culture by boiled DNA extraction method using commercial DNA extraction kit, QIAamp DNA Mini Kit (QIAGEN, Hilden, Germany). Quality and quantity of the extracted DNA were measured using a NanoDrop ND-2000 spectrophotometer (Thermo Fisher Scientific, Waltham, MA). DNA extracts with A260/280 and A260/230 ratios

**Table 1. List of primers used in this study.**

| Name of Primers | Targeted gene | Primer sequences (5′-3′) | Amplicon size (bp) | References |
|---|---|---|---|---|
| *malB* (F) | *malB* | 5′GACCTCGGTTTAGTTCACAGA3′ | 585 | [30] |
| *malB* (R) | | 5′ CACACGCTGACGCTGACCA3′ | | |
| *tetA* (F) | *tetA* | 5′ GGTTCACTCGAACGACGTCA3′ | 577 | [41] |
| *tetA* (R) | | 5′ CTGTCCGACAAGTTGCATGA3′ | | |
| *blaTEM* (F) | *blaTEM* | 5′ CATTTCCGTGTCGCCCTTAT3′ | 793 | |
| *blaTEM* (R) | | 5′ TCCATAGTTGCCTGACTCCC3′ | | |

of ∼ 1.80 and 2.00 to 2.20, respectively, were considered as high-purity DNA samples [38] and stored at -20˚C prior to PCR amplification [37,39]. Amplification of targeted DNA was carried out in a 20 μL reaction mixture, which included 3 μL nuclease-free water, 10 μL 2X master mixture (Promega, Madison, WI, USA), 1 μL each of forward and reverse primers, and 5 μL DNA template. PCR-positive controls consisted of *E. coli* genomic DNA previously confirmed for the target genes [37]. PCR-negative controls utilized non-template controls with PBS instead of genomic DNA. The amplified PCR products were then subjected to electrophoresis on a 1.5% agarose gel and visualized using an ultraviolet transilluminator (Biometra, Gottingen, Germany). A 100 bp DNA ladder (Promega, Madison, WI, USA) was used to validate the expected sizes of the amplified PCR products [37,40]. Finally, 85 isolates were confirmed as *E. coli* through species-specific PCR.

## 2.3 Antimicrobial susceptibility assay

The antimicrobial susceptibility profiles of 54 randomly selected *E. coli* isolates (out of 85 confirmed isolates) were assessed using the disk diffusion test (DDT), in accordance with the guidelines outlined in the Clinical Laboratory Standards Institute (CLSI) 2023 (M100 33rd Edition) [42]. Eleven antibiotics from nine commonly practiced antibiotic classes in Bangladesh were employed. These were ciprofloxacin (CIP, 5 μg), gentamicin (GEN, 10 μg), tetracycline (TET, 30 μg), ceftriaxone (CTR, 30 μg), ampicillin (AMP, 25 μg), ceftazidime (CAZ, 5 μg), chloramphenicol (C, 30 μg), imipenem (IMP, 10 μg), fosfomycin (FOS, 50 μg), nitrofurantoin (NIT, 300 μg), and cotrimoxazole (COT, 25 μg). The isolated colonies were taken into 4–5 mL of nutrient broth for performing DDT. After preparing the broth cultures, isolates were incubated for 4–5 hrs at 37˚C, and the turbidity of bacterial suspensions was adjusted with the 0.5 McFarland unit (HiMedia, India). After that, the dried surface of a Muller Hilton (MH) agar plate was inoculated by spreading the broth suspension on the surface with sterile cotton swabs. Finally, the antibiotic disks were applied on the surface of the agar plates, and left for overnight (>16 hrs.) incubation at 37˚C. The isolates were categorized as susceptible, intermediate, and resistant according to CLSI guidelines [42]. Multidrug resistance (MDR) patterns, defined as resistance to ≥ 3 antibiotics, were identified using the protocol outlined by Saha et al. and Sultana et al. [43,44]. The Multiple Antibiotic Resistance (MAR) index was calculated by dividing the number of antibiotics to which an isolate was resistant by the total number of antibiotics tested [45]. *E. coli* strain ATCC25922 was used as the negative control in the antimicrobial susceptibility tests.

## 2.4 Molecular detection of antibiotic-resistant genes in *E. coli*

To detect antibiotic-resistant genes in the *E. coli* isolates (n = 54), simplex PCR assays were conducted for beta-lactamase genes (e.g., *blaTEM*) (S3A Fig) and tetracycline resistance genes (e.g., *tetA*) (S3B Fig) using specific primers (Table 1). For both genes, PCR was conducted

with a final volume of 20 μL. The PCR conditions for the *tetA* gene involved an initial denaturation at 95˚C for 5 min, followed by 32 cycles of denaturation at 95˚C for 1 min, annealing at 57˚C for 1 min, and extension at 72˚C for 1 min. A final extension step was carried out at 72˚C for 10 min. For the *blaTEM* gene, the thermal profile included an initial denaturation at 95˚C for 1 min, followed by 34 cycles of denaturation at 95˚C for 1 min, annealing at 56˚C for 1 min, and extension at 72˚C for 1 min, with a final extension at 72˚C for 7 min [40]. Although a positive control was not included for the resistance genes, a non-template control (NTC), which contained no DNA, was used to ensure the absence of contamination.

## 2.5 Statistical analysis

Data were entered into Microsoft Excel 2020® (Microsoft Corp., Redmond, WA, USA) and analyzed using SPSS version 25 (IBM Corp., Armonk, NY, USA) and GraphPad Prism version 8.4.3 (GraphPad Software, Inc.). The Pearson's chi-square test was conducted to compare the occurrence of *E. coli* across different sample categories (e.g., DNCC, DSCC, and GCC). Prevalence percentages were calculated by dividing the number of positive samples in each category by the total number of samples tested within that category [46,47]. The prevalence formula was applied for determining occurrence percentage of *E. coli*. The AMR patterns, resistance, intermediate and sensitivity, and MAR index were calculated using the CLSI (2023) guideline using the cut-off as provided in the brochure of the manufacturer (Liofilchem®, Italy). Additionally, an identical test was done to determine whether the presence of resistance genes caused variations in phenotypic antibiotic resistance. For the test, p < 0.05 was considered statistically significant.

## Results

### 3.1 Overall prevalence of *E. coli*

In this study, 145 samples were collected from DNCC, DSCC, and GCC and subjected to analysis. The identification process involved culturing the samples, performing biochemical tests, and conducting PCR targeting the *malB* gene (**S2 Fig**). Out of the total samples, 85 isolates were confirmed as *E. coli*. This resulted in an overall prevalence of *E. coli* in the studied samples of 58.62% (95% CI: 50.48–66.31) (**S2 Table**). This prevalence indicates that more than half of the samples contained *E. coli*, reflecting its significant presence in the studied areas.

### 3.2 Prevalence of *E. coli* in urban (DNCC and DSCC) areas of Bangladesh

In DNCC, a total of 85 samples, including vegetables, water, and soil, were collected from rooftop (n = 43) and surface (n = 42) gardens. The overall prevalence of *E. coli* in these samples was 44.70% (95% CI, 34.59–55.28) (**Fig 1A, S3 Table**). However, the prevalence of *E. coli* was lower in rooftop gardens, at 20.93% (95% CI: 11.42–35.20), compared to surface gardens, which had a higher occurrence of 69.04% (95% CI: 53.97–80.92) (**Fig 1B, S4 Table**). In the rooftop samples of DNCC, *E. coli* was detected in vegetables and soil with frequencies of 28.0% (95% CI: 14.28–47.57) and 25% (95% CI: 4.44–59.07), respectively. Water samples from the rooftop gardens were found to be free of *E. coli* (**Fig 1C, S5 Table**). In contrast, surface samples showed *E. coli* prevalence rates of 86.95% (95% CI: 67.87–95.46) in vegetables, 12.5% (95% CI: 0.64–47.08) in water, and 72.73% (95% CI: 43.43–90.25) in soil (**Fig 1C, S6 Table**).

Similarly, from DSCC, 30 samples were collected, including rooftop gardens (n = 15) and surface gardens (n = 15), with an 80.0% prevalence found (95% CI, 62.69–90.49) (**Fig 1A, S3 Table**). Consistent with DNCC, the prevalence of *E. coli* was lower in rooftop gardens (73.33%, 95% CI, 48.05–89.10) compared to surface gardens (86.67%, 95% CI, 62.12–97.63) (**Fig 1B, S4 Table**). In rooftop gardens of DSCC, *E. coli* was found in 70% of vegetable samples

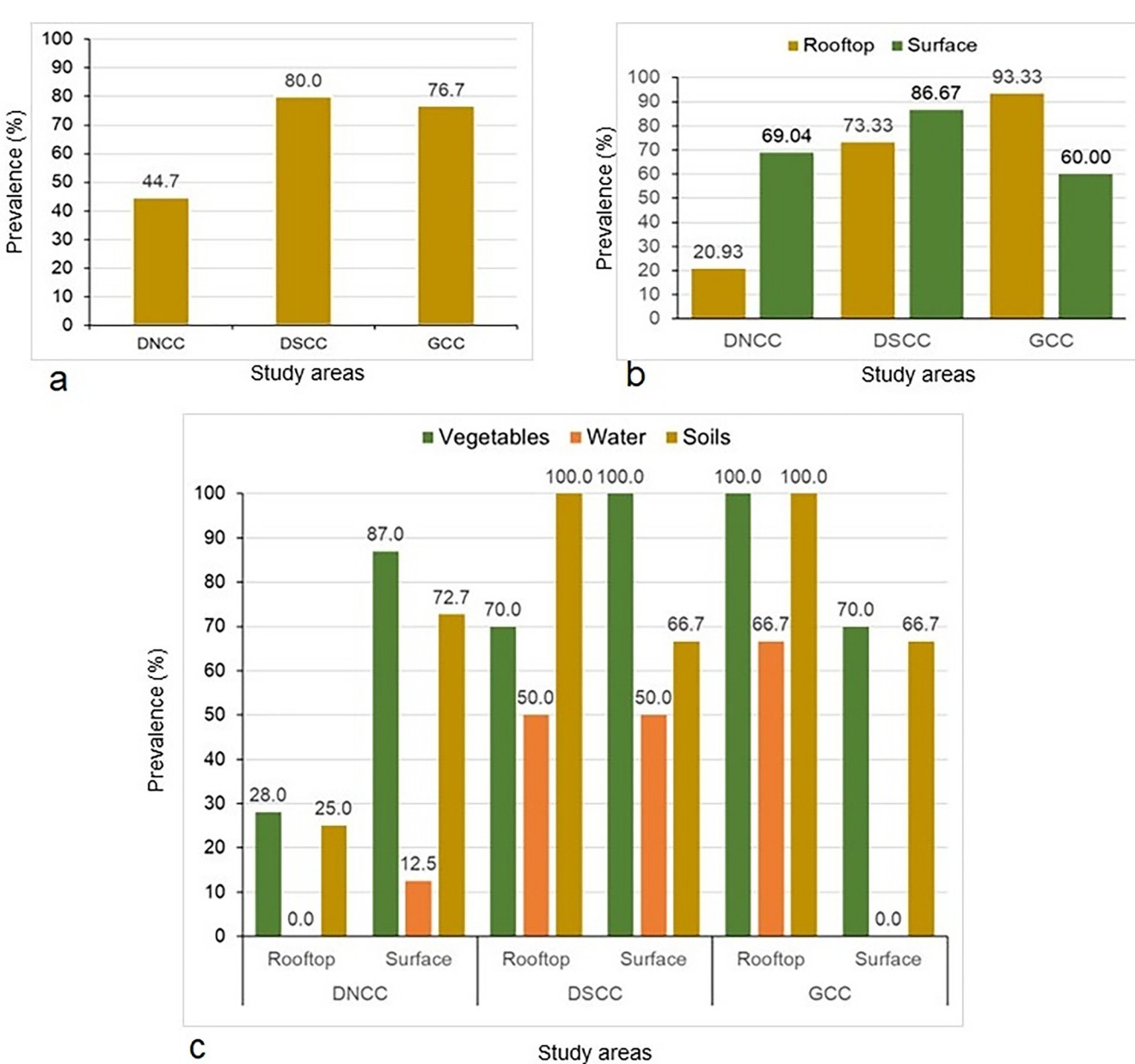

**Fig 1. Prevalence of *E. coli* based on study areas (DNCC, DSCC, and GCC), locations (rooftop and surface gardens), and sample types (vegetables, water and soils).**

(95% CI, 39.67–89.22), 50% of water samples (95% CI, 2.56–97.43), and 100% of soil samples (95% CI, 43.85–100) (**Fig 1C**, **S5 Table**). Conversely, in surface gardens of DSCC, *E. coli* prevalence was highest in vegetables (100%, 95% CI, 72.24–100), followed by soil (66.67%, 95% CI, 11.84–98.29) and water samples (50%, 95% CI, 2.56–97.43) (**Fig 1C**, **S6 Table**).

### 3.3 Prevalence of *E. coli* in peri-urban (GCC) areas of Bangladesh

In the peri-urban area of GCC, a total of 30 samples (15 from rooftop gardens and 15 from surface gardens) were collected and analyzed, with an overall *E. coli* prevalence of 76.7% (95% CI,

59.07–44.20) (**Fig 1A, S3 Table**). In contrast to the urban areas (DNCC and DSCC), *E. coli* prevalence was higher in rooftop gardens (93.33%, 95% CI, 70.18–99.65) compared to surface gardens (60.0%, 95% CI, 35.74–80.17) in the peri-urban area of GCC (**Fig 1B, S4 Table**). Specifically, *E. coli* was found in 100% of vegetable (95% CI, 72.24–100) and soil (95% CI, 17.76–100) samples from rooftop gardens, while water samples had a prevalence of 66.7% (95% CI, 11.84–98.29) (**Fig 1C, S5 Table**). In contrast, water samples from surface gardens were free of *E. coli*. However, *E. coli* was detected in 70.0% of vegetable (95% CI, 39.68–89.22) and 66.7% of soil (95% CI, 11.84–98.29) samples from the same gardens (**Fig 1C, S6 Table**).

### 3.4 Antibiogram profile of *E. coli*

The overall antibiogram profile of isolated *E. coli* is presented in **Fig 2**. Out of the 85 isolates, a random selection of 54 was subjected to antibiogram testing. Resistance was observed across all isolates to ampicillin (AMP; 100%), with varying resistance rates noted for ciprofloxacin (CIP; 25.92%), tetracycline (TET; 14.81%), cotrimoxazole (COT; 14.81%), imipenem (IMP; 9.25%), and fosfomycin (FOS; 1%) (**Fig 2**). Additionally, these isolates showed intermediate resistance to ciprofloxacin (CIP; 74.0%), imipenem (IMP; 37.0%), and fosfomycin (FOS; 33.0%). Fortunately, the tested *E. coli* isolates were 100% susceptible to gentamicin (GEN), ceftazidime (CAZ), chloramphenicol (C), nitrofurantoin (NIT), and ceftriaxone (CTR). They

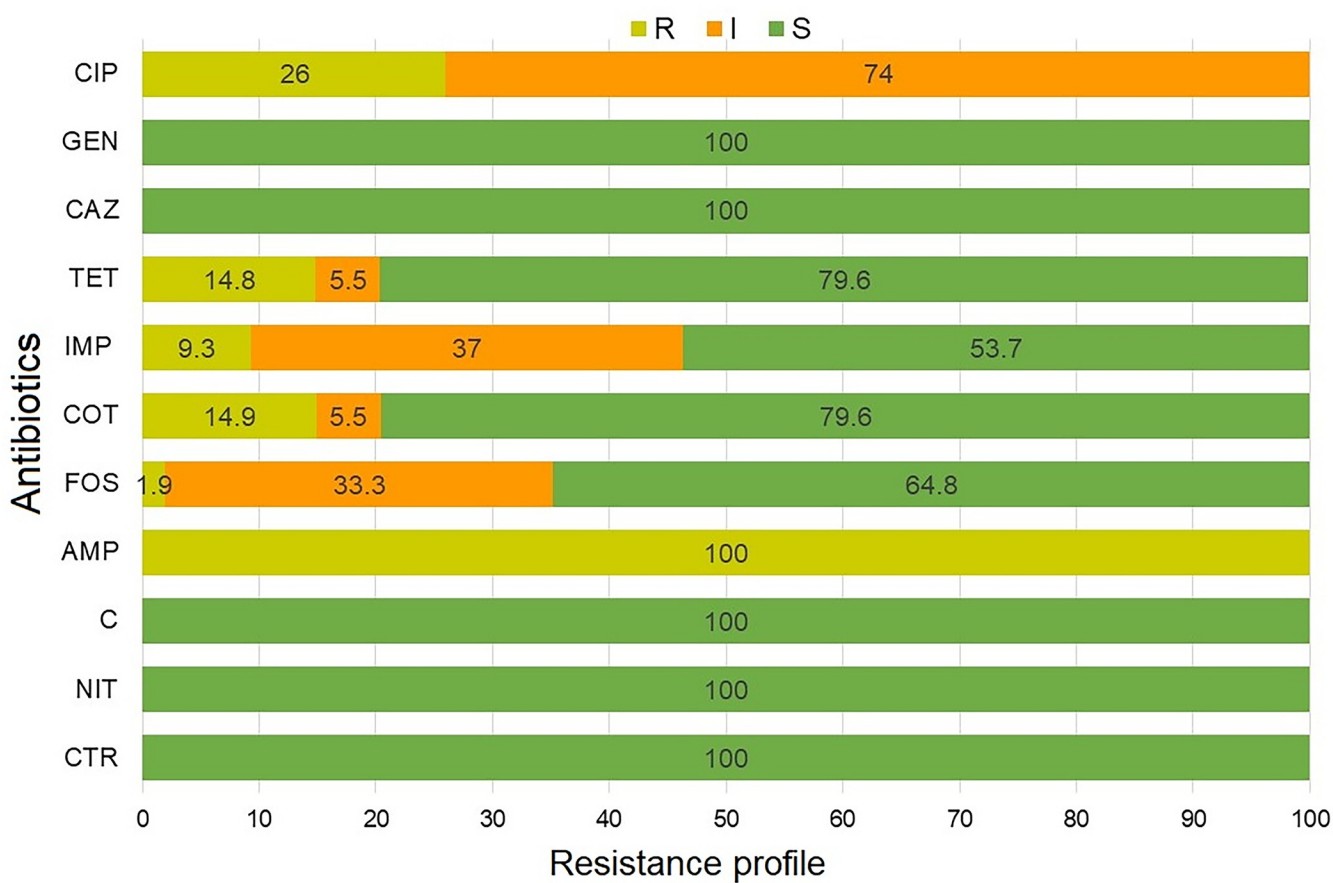

**Fig 2. Overall resistance rates of the 54 *E. coli* isolates to 11 antibiotics.** The percentage of R (Resistant, olive), I (Intermediate resistant, orange), and S (Susceptible, green) profiles are indicated for each antibiotic inside the bar chart. CIP: Ciprofloxacin, GEN: Gentamicin, CAZ: Ceftazidime, TET; Tetracycline, IMP; Imipenem, COT; Cotrimoxazole, FOS; Fosfomycin, AMP; Ampicillin, C; Chloramphenicol, NIT; Nitrofurantoin and CTR: Ceftriaxone.

were also susceptible to tetracycline (TET; 79.6%), cotrimoxazole (COT; 79.6%), fosfomycin (FOS; 64.8%), and imipenem (IMP; 53.7%) (Fig 2). However, bivariate analysis of the tested antibiotics revealed a strong positive and significant correlation between resistance to tetracycline and cotrimoxazole (p = 0.002, ρ = 0.413) (Table 2).

### 3.5 Phenotypic resistance patterns of the multidrug resistance *E. coli* isolates

Table 3 presents the phenotypic multidrug resistance (MDR) patterns of the *E. coli* isolates. Among the 54 isolates, 48.14% (95% CI: 35.39–61.14) exhibited MDR. In total, 10 distinct antibiotic resistance patterns were identified. The most prevalent pattern was pattern no. 1 (AMP, TET, CIP, COT, IMP), observed in 14.81% of isolates, followed by pattern no. 2(AMP, TET, COT) in 12.96% of the isolates. Patterns no. 3, 4, 5, 6, and 7, which include combinations like (AMP, TET, CIP), (AMP, CIP, COT), (AMP, COT), (AMP, TET), and (AMP, IMP) showed a prevalence of 11.11%. The least prevalent pattern was pattern no. 8 (AMP, FOS), 9 (AMP, CIP), and 10 (AMP), observed in 5.55% of the isolates. The multiple antibiotic resistance (MAR) indices were found to vary between 0.09 and 0.45 (Table 3).

### 3.6 Genotypic resistance patterns of *E. coli* isolates

The presence of two AMR genes (e.g., *blaTEM* and *tetA*) in all *E. coli* isolates was assessed by PCR (Fig 3). Among the 54 randomly selected *E. coli* isolates, all 54 (100.0%) exhibited phenotypic resistance to ampicillin. In contrast, only 8 isolates (14.81%) demonstrated phenotypic resistance to tetracycline. In the *E. coli* isolates that exhibited resistance to ampicillin, the *blaTEM* gene was detected in 81.48% (95% CI: 69.16–89.61, 44 out of 54 isolates). In those resistant to tetracycline, the *tetA* gene was present in 25.0% (95% CI: 4.44–59.07, 2 out of 8 isolates) (Fig 3).

### Discussion

Foodborne illnesses have increasingly become a significant concern across communities globally [48,49]. This rise in foodborne illnesses is attributed to variations in distribution patterns, manufacturing processes, and consumer behaviors [11]. This study on the prevalence and

**Table 2. Pearson correlation coefficient to assess the pairs of any of two resistant antibiotics used in *E. coli*.**

| Antibiotics | | TET | CIP | COT | FOS | IMP |
|---|---|---|---|---|---|---|
| TET | Pearson Correlation | 1 | | | | |
| | Sig. (2-tailed) | | | | | |
| AMP | Pearson Correlation | -0.057 | | | | |
| | Sig. (2-tailed) | 0.681 | | | | |
| CIP | Pearson Correlation | 0.229 | 1 | | | |
| | Sig. (2-tailed) | 0.096 | | | | |
| COT | Pearson Correlation | 0.413** | 0.229 | 1 | | |
| | Sig. (2-tailed) | 0.002 | 0.096 | | | |
| FOS | Pearson Correlation | -0.057 | -0.081 | -0.057 | 1 | |
| | Sig. (2-tailed) | 0.681 | 0.559 | 0.681 | | |
| IMP | Pearson Correlation | 0.226 | 0.103 | 0.226 | -0.044 | 1 |
| | Sig. (2-tailed) | 0.1 | 0.46 | 0.1 | 0.753 | |

** Correlation is significant at the 0.01 level (2-tailed). TET: Tetracycline, CIP: Ciprofloxacin, COT: cotrimoxazole, FOS: Fosfomycin, IMP: Imipenem.

**Table 3. Resistance patterns of multidrug resistant (MDR) *E. coli* isolates.**

| Pattern No. | Resistance patterns | No. of antibiotics (classes) | No. of MDR Isolates | MDR (%) | MAR index |
|---|---|---|---|---|---|
| 1 | AMP, TET, CIP, COT, IMP | 5(5) | 7 | 48.14 | 0.45 |
| 2 | AMP, TET, COT | 3(3) | 7 | | 0.27 |
| 3 | AMP, TET, CIP | 3(3) | 6 | | |
| 4 | AMP, CIP, COT | 3(3) | 6 | | |
| 5 | AMP, COT | 2(2) | 7 | | 0.18 |
| 6 | AMP, TET | 2(2) | 6 | | |
| 7 | AMP, IMP | 2(2) | 6 | | |
| 8 | AMP, FOS | 2(2) | 3 | | |
| 9 | AMP, CIP | 2(2) | 3 | | |
| 10 | AMP | 1(1) | 3 | | 0.09 |

TET: Tetracycline, AMP: Ampicillin, CIP: Ciprofloxacin, COT: Cotrimoxazole, FOS: Fosfomycin, IMP: Imipenem.

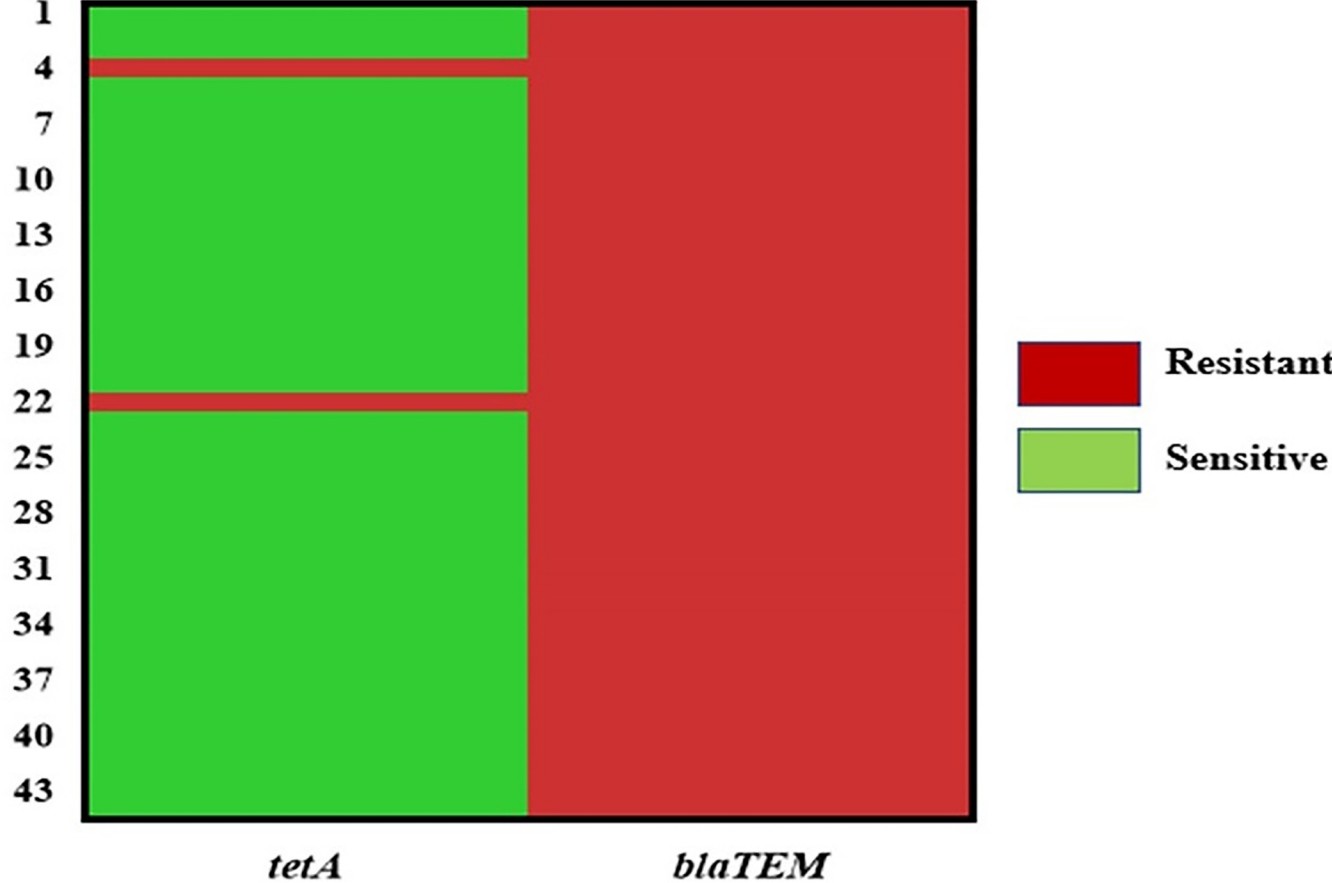

**Fig 3. Heatmap illustrating the distribution of resistance genes (*blaTEM* and *tetA*) in *E. coli* isolates, with the X-axis representing the resistance genes and the Y-axis displaying the isolates.** In the heatmap, the red color denotes resistant isolates, while the green color indicates sensitive isolates.

antibiotic resistance profiles of *E. coli* in urban (DNCC and DSCC) and peri-urban (GCC) rooftop and surface gardens explored the frequency of this pathogen in these environments and its resistance to various antibiotics. We also assessed how often *E. coli* is found in rooftop versus surface gardens and examined the antibiotic resistance patterns of isolated strains and the presence of specific resistance genes. In this study, the overall prevalence of *E. coli* was found to be 58.62%, indicating a significant presence of this pathogen in the surveyed urban and peri-urban gardens. In a similar study, Nipa et al. reported a 40.62% prevalence of *E. coli* in fresh salad vegetables, which is lower compared to the prevalence observed in the current study [50]. Raw vegetables are particularly vulnerable to contamination by pathogenic bacteria like *E. coli*, which can either be dispersed on the plant surface or embedded as microcolonies within plant tissues [51]. In developing countries like Bangladesh, the incidence of foodborne illnesses linked to contaminated vegetables is notably high [52,53]. The lack of research and surveillance often results in many outbreaks going unreported, with only a limited number documented in scientific literature.

The study provides a detailed examination of *E. coli* prevalence in urban gardens, revealing distinct patterns between rooftop and surface gardens in DNCC and DSCC. In DNCC, the overall *E. coli* prevalence was 44.70%, with rooftop gardens exhibiting a lower prevalence (20.93%) compared to surface gardens (69.04%). Specifically, *E. coli* was present in 28.0% of rooftop vegetable samples and 25% of rooftop soil samples, while no *E. coli* was detected in rooftop water samples. Conversely, surface gardens had a significantly higher prevalence of *E. coli* in vegetables (86.95%), with lower levels in water (12.5%) and a substantial prevalence in soil (72.73%). The findings from DSCC corroborate these trends, showing an overall *E. coli* prevalence of 80.0%. Rooftop gardens in DSCC had a prevalence of 73.33%, whereas surface gardens had a higher prevalence of 86.67%. In rooftop gardens, *E. coli* was found in 70% of vegetable samples, 50% of water samples, and 100% of soil samples. In surface gardens, the pathogen was detected in 100% of vegetable samples, 66.67% of soil samples, and 50% of water samples. However, in the peri-urban area of GCC, a total of 30 samples from rooftop and surface gardens revealed a high overall *E. coli* prevalence. Rooftop gardens had a higher prevalence compared to surface gardens. Specifically, *E. coli* was found in 100% of vegetable and soil samples from rooftop gardens and in 66.7% of water samples. In surface gardens, *E. coli* was present in 70.0% of vegetable samples and 66.7% of soil samples but was absent from water samples. These results indicate that rooftop gardens experience more extensive contamination, particularly in vegetables and soil, highlighting the need for enhanced sanitation and management practices in both garden types to address *E. coli* contamination [11,24]. Agricultural practices in rooftop and surface gardens in urban and peri-urban areas of Bangladesh, such as using manure-based fertilizers, irrigating with possibly contaminated water, and applying pesticides or antibiotics, may introduce and propagate MDR bacteria. This contamination can transfer to plants and soil, posing public health risks when these vegetables are consumed. Although this study analyzed a limited sample size, it marks, to the best of our knowledge, the first investigation of antimicrobial resistance in urban and peri-urban garden systems in Bangladesh, highlighting a critical area for further research and monitoring.

While the presence of *E. coli* in these urban settings has been shown not to be a good indicator of pathogens, we assume that *E. coli* is prevalent in urban and rooftop gardens due to factors like contaminated water sources, exposure to animal waste, insufficient hygiene practices, and soil quality issues [54]. These conditions create ideal environments for bacterial contamination, impacting food safety. The elevated *E. coli* prevalence in surface gardens may stem from differences in soil type, irrigation, fertilization, and exposure to human or animal activity, which vary significantly from rooftop gardens [55,56]. The role of soil as a primary reservoir for *E. coli* suggests its capacity to sustain and spread the bacterium to plants and ultimately to

humans [57,58]. Additionally, the use of unsanitized organic fertilizers or compost in surface gardens can introduce bacteria, while rooftop gardens exhibited no contamination in water samples, implying more controlled inputs. Lower prevalence in water compared to vegetables may indicate dilution effects or sampling differences, while high soil prevalence reinforces soil's potential as an *E. coli* reservoir [58]. This underscores the public health significance of soil in urban and rooftop gardens, where contaminated soil can directly impact food safety and increase the risk of *E. coli*-related infections.

The antibiogram of 54 *E. coli* isolates showed universal resistance to ampicillin (100.0%) and varying resistance rates to ciprofloxacin, tetracycline, cotrimoxazole, imipenem, and Fosfomycin ($< 30.0\%$). Intermediate resistance was noted for ciprofloxacin (74%), imipenem (37%), and fosfomycin (33%). There were high susceptibility rates (100.0%) among the *E. coli* isolates to gentamicin, ceftazidime, chloramphenicol, nitrofurantoin, and ceftriaxone. Remarkably, a significant correlation was found between resistance to tetracycline and cotrimoxazole ($p = 0.002$, $\rho = 0.413$). In this study, 48.14% of the *E. coli* isolates showed multidrug resistance (MDR), with 10 distinct resistance patterns identified. The most common pattern involved ampicillin alone, while other patterns included combinations of ampicillin with various antibiotics. These results moderately contrast with those of Cao et al., who reported a 92.9% multidrug resistance rate among *E. coli* isolates from retail fresh vegetables in Shaanxi Province, China [59]. This study also noted variability in the MAR indices among the isolates. Antibiotic-resistant bacteria like *E. coli* can migrate from one location to another and from the environment to humans via the consumption of raw vegetables. This transmission pathway underscores the importance of monitoring and managing antibiotic resistance in agricultural and food safety practices [60]. Unlike our findings, which showed 14.81% of *E. coli* isolates resistant to tetracycline, two previous studies reported higher resistance rates of 80% and 43.06%, respectively [61,62]. Many studies have documented the presence of drug-resistant *E. coli* and other coliforms in vegetables [53,60,63]. This highlights a concerning trend in food safety, as the consumption of contaminated vegetables can facilitate the spread of antibiotic-resistant bacteria to humans. The fact that ampicillin is a clinical antibiotic makes this more disturbing and suggests the source of contamination may have been from human waste and thus corroborate the assumption that contamination is due to discharge from anthropogenic sources [53]. This result also agrees with a recent report of high tetracycline resistance observed among *E. coli* isolates [53]. Moreover, the detection of resistance phenotype was also supported by the detection of *blaTEM* and *tetA* genes in the *E. coli* isolates. All 54 isolates were resistant to ampicillin, with 81.48% carrying the *blaTEM* gene. However, only 14.81% showed resistance to tetracycline, and among these, 25.0% had the *tetA* gene. Statistical analysis indicated that while ampicillin-resistant isolates had similar frequencies of *blaTEM* and *tetA* genes, tetracycline-resistant isolates had a significantly ($p = 0.035$) higher prevalence of *tetA* compared to *blaTEM*, highlighting differences in phenotypic resistance. *E. coli* is a common component of the intestinal flora and is generally harmless. However, antibiotic resistance genes like *blaTEM* and *tetA* present in commensal *E. coli* can be transferred to pathogenic strains like *E. coli* O157 or *Salmonella* spp. [53,64]. This gene transfer can lead to serious health issues, complicating treatment and increasing the risk of severe infections [64]. In contrast to our findings, Kim and Woo (2014) characterized antimicrobial-resistant *E. coli* from organic vegetables and found a lower prevalence of *blaTEM* genes (3.6%) but a higher prevalence of *tetA* genes (10.7%) [63].

Industrialization has increased health awareness, leading people to prefer homegrown vegetables with minimal pesticides, fertilizers, and antibiotics. Consequently, antibiotic resistance remains moderate, with multidrug resistance (14.81%) in *E. coli* being a concern. Sustainable farming practices, regular hygiene, and farm management are essential to control resistance.

Effective management is crucial to mitigate the risks posed by antibiotic-resistant *E. coli* in urban agriculture. While *E. coli* can be found in various environments, the lack of detailed information regarding gardening practices in the three areas limits the applicability of our findings for effective garden management. To enhance vegetable safety for human consumption, further research should focus on comparing soil quality, water sources, and preparation techniques. This would provide actionable insights for improving hygiene practices and mitigating contamination risks in both rooftop and surface gardens.

## 5. Conclusion

MDR *E. coli* poses a significant public health threat worldwide. The findings from the present study provide the prevalence of *E. coli* in vegetables, water and soil samples at the urban (DNCC and DSCC) and peri-urban (GCC) rooftop and surface gardens harboring *blaTEM* and *tet*A resistance genes. Detection was confirmed both phenotypically and genotypically via PCR, raising serious public health concerns. Fresh salad vegetables could be a potential source of drug-resistant *E. coli*. This study is the first in Bangladesh to report MDR *E. coli* from rooftop vegetables, soil, and water in urban (DNCC and DSCC) and peri-urban (GCC) rooftop and surface gardens of Bangladesh. Overall, these findings emphasize the importance of monitoring and managing *E. coli* contamination in urban and peri-urban gardens, especially in areas with high prevalence. Our findings underscore the importance of public awareness about hygiene practices and environmental controls to minimize contamination risks in both surface and rooftop gardens. Encouraging regular monitoring and thoroughly washing rooftop garden produce with safe, potable water is essential to safeguard public health and prevent potential foodborne illnesses. Regular training and awareness programs for gardeners about best practices can also enhance overall food safety and reduce public health risks.

## Supporting information

**S1 Table. Sampling information of the study.**
(DOCX)

**S2 Table. Number of *E. coli* positive samples from the study areas.**
(DOCX)

**S3 Table. Prevalence of *E. coli* in the study areas.**
(DOCX)

**S4 Table. Prevalence of *E. coli* in the rooftop and surface gardens of the study areas.**
(DOCX)

**S5 Table. Prevalence of *E. coli* in different samples of rooftop gardens.**
(DOCX)

**S6 Table. Prevalence of *E. coli* in different samples of surface gardens.**
(DOCX)

**S1 Fig. Study areas and sampling locations.** (a) Urban (Dhaka North City Corporation; DNCC and Dhaka South City Corporation; DSCC) and peri-urban (Gazipur City Corporation; GCC) areas of Bangladesh. (b) Rooftop gardens and (c) Surface gardens.
(JPG)

**S2 Fig. PCR amplification of *malB* gene of *Escherichia coli*.** Lane 1: 1 kb DNA Marker; Lane 2: Negative control; Lane 3: Positive control; and Lane 4–13: Representative *E. coli* isolates.
(JPG)

**S3 Fig.** (a) PCR amplification of beta-lactamase-producing *blaTEM* gene in representative *E. coli* isolates. (b) PCR amplification of tetracycline resistance *tetA* gene in representative *E. coli* isolates.
(JPG)

## Acknowledgments

The authors would like to thank the authority who provided us with the samples from diverse environment to the support the research.

## Author Contributions

**Conceptualization:** M. Nazmul Hoque, Md. Tanvir Rahman.

**Data curation:** Pritom Kumar Pramanik, M. Nazmul Hoque, Md. Liton Rana, Md. Saiful Islam, Md. Ashek Ullah, Fahim Haque Neloy.

**Formal analysis:** Pritom Kumar Pramanik, M. Nazmul Hoque, Md. Liton Rana, Md. Saiful Islam, Md. Ashek Ullah, Fahim Haque Neloy.

**Funding acquisition:** Srinivasan Ramasamy, Pepijn Schreinemachers, Ricardo Oliva, Md. Tanvir Rahman.

**Investigation:** Pritom Kumar Pramanik, Md. Liton Rana, Pepijn Schreinemachers, Ricardo Oliva.

**Methodology:** Pritom Kumar Pramanik, M. Nazmul Hoque, Md. Liton Rana.

**Project administration:** Srinivasan Ramasamy, Pepijn Schreinemachers, Ricardo Oliva, Md. Tanvir Rahman.

**Resources:** Srinivasan Ramasamy, Pepijn Schreinemachers, Ricardo Oliva, Md. Tanvir Rahman.

**Software:** Md. Tanvir Rahman.

**Supervision:** Md. Tanvir Rahman.

**Validation:** M. Nazmul Hoque.

**Visualization:** Pritom Kumar Pramanik, M. Nazmul Hoque.

**Writing – original draft:** Pritom Kumar Pramanik, Md. Liton Rana, Md. Saiful Islam, Md. Ashek Ullah, Fahim Haque Neloy.

**Writing – review & editing:** M. Nazmul Hoque, Srinivasan Ramasamy, Pepijn Schreinemachers, Ricardo Oliva, Md. Tanvir Rahman.

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
