## [Decision Letter · Decision Letter 0]

21 Oct 2024

PONE-D-24-32199Prevalence and antibiotic resistance of Escherichia coli in urban and peri-urban garden ecosystems in BangladeshPLOS ONE

Dear Dr. Rahman,

Thank you for submitting your manuscript to PLOS ONE. After careful consideration, we feel that it has merit but does not fully meet PLOS ONE’s publication criteria as it currently stands. Therefore, we invite you to submit a revised version of the manuscript that addresses the points raised during the review process.

We look forward to receiving your revised manuscript.

Kind regards,

Bilal Aslam, PhD

Academic Editor

PLOS ONE

Journal requirements: When submitting your revision, we need you to address these additional requirements. 1. Please ensure that your manuscript meets PLOS ONE's style requirements, including those for file naming. The PLOS ONE style templates can be found at https://journals.plos.org/plosone/s/file?id=wjVg/PLOSOne_formatting_sample_main_body.pdf and https://journals.plos.org/plosone/s/file?id=ba62/PLOSOne_formatting_sample_title_authors_affiliations.pdf 2. PLOS ONE now requires that authors provide the original uncropped and unadjusted images underlying all blot or gel results reported in a submission’s figures or Supporting Information files. This policy and the journal’s other requirements for blot/gel reporting and figure preparation are described in detail at https://journals.plos.org/plosone/s/figures#loc-blot-and-gel-reporting-requirements and https://journals.plos.org/plosone/s/figures#loc-preparing-figures-from-image-files. When you submit your revised manuscript, please ensure that your figures adhere fully to these guidelines and provide the original underlying images for all blot or gel data reported in your submission. See the following link for instructions on providing the original image data: https://journals.plos.org/plosone/s/figures#loc-original-images-for-blots-and-gels.   In your cover letter, please note whether your blot/gel image data are in Supporting Information or posted at a public data repository, provide the repository URL if relevant, and provide specific details as to which raw blot/gel images, if any, are not available. Email us at plosone@plos.org if you have any questions. 3. In your Methods section, please provide additional information regarding the permits you obtained for the work. Please ensure you have included the full name of the authority that approved the field site access and, if no permits were required, a brief statement explaining why. 4. We suggest you thoroughly copyedit your manuscript for language usage, spelling, and grammar. If you do not know anyone who can help you do this, you may wish to consider employing a professional scientific editing service.  The American Journal Experts (AJE) (https://www.aje.com/) is one such service that has extensive experience helping authors meet PLOS guidelines and can provide language editing, translation, manuscript formatting, and figure formatting to ensure your manuscript meets our submission guidelines. Please note that having the manuscript copyedited by AJE or any other editing services does not guarantee selection for peer review or acceptance for publication.  Upon resubmission, please provide the following: The name of the colleague or the details of the professional service that edited your manuscript A copy of your manuscript showing your changes by either highlighting them or using track changes (uploaded as a *supporting information* file) A clean copy of the edited manuscript (uploaded as the new *manuscript* file)”. 5. We note that the grant information you provided in the ‘Funding Information’ and ‘Financial Disclosure’ sections do not match.  When you resubmit, please ensure that you provide the correct grant numbers for the awards you received for your study in the ‘Funding Information’ section. 6. Thank you for stating the following financial disclosure:  [This work was conducted as part of the CGIAR Research Initiative on Resilient Cities Through Sustainable Urban and Peri-urban Agri-food Systems and is supported by contributors to the CGIAR Trust Fund (https://www.cgiar.org/funders).].  Please state what role the funders took in the study.  If the funders had no role, please state: ""The funders had no role in study design, data collection and analysis, decision to publish, or preparation of the manuscript."" If this statement is not correct you must amend it as needed. Please include this amended Role of Funder statement in your cover letter; we will change the online submission form on your behalf. 7. Please provide a complete Data Availability Statement in the submission form, ensuring you include all necessary access information or a reason for why you are unable to make your data freely accessible. If your research concerns only data provided within your submission, please write "All data are in the manuscript and/or supporting information files" as your Data Availability Statement.

Reviewers' comments:

Reviewer's Responses to Questions

**Comments to the Author**

1. Is the manuscript technically sound, and do the data support the conclusions?

Reviewer #1: Yes

Reviewer #2: Yes

Reviewer #3: Yes

Reviewer #4: Partly

2. Has the statistical analysis been performed appropriately and rigorously? 

Reviewer #1: Yes

Reviewer #2: No

Reviewer #3: Yes

Reviewer #4: Yes

3. Have the authors made all data underlying the findings in their manuscript fully available?

Reviewer #1: Yes

Reviewer #2: Yes

Reviewer #3: Yes

Reviewer #4: Yes

4. Is the manuscript presented in an intelligible fashion and written in standard English?

Reviewer #1: Yes

Reviewer #2: Yes

Reviewer #3: Yes

Reviewer #4: Yes

5. Review Comments to the Author

Reviewer #1: The manuscript entitled:’” Prevalence and antibiotic resistance of Escherichia coli in urban and peri-urban garden ecosystems in Bangladesh” is a nicely written well-designed study. AMR is a major health problem across the globe. Here the authors have focused MDR E. coli in various gardening systems in Bangladesh. Methodologies are reproducible along with detailed results and discussion. Vegetables grown in rooftop gardens could be potential sources for MDR E. coli. As the authors have mentioned it is the first such study in Bangladesh describing MDR E. coli in rooftop vegetables, soil, and water in urban and peri-urban rooftop and surface gardens of Bangladesh. I believe the findings of the study could be considered in developing guidelines for better rooftop garden management in Bangladesh for better public health linked to MDR E. coli.

Nevertheless, I have a few comments as follows:

Please write the function of malB gene used to detect E.coli.

Name the vegetables under methodology section, scientific name…

What was the basis of selecting those antibiotics for the sensitivity test?

Please mention the year of CLSI in the methodology section, 2022?2023/2024??

What could be is the explanation for observing more E. coli in the surface garden than rooftop garden? Is it expected??

Water samples from the rooftop gardens were found negative for E. coli, any speculation?

There are several tet family genes, why in this study only tetA gene primer was used for genotype.

Woo (2014)found a lower prevalence of blaTEM genes (3.6%) but a higher prevalence of tetA genes (10.7%), just the opposite of your study, what could be the reason??

Mention the major limitation of the study.

Reviewer #2: The manuscript entitled “Prevalence and antibiotic resistance of Escherichia coli in urban and peri-urban garden ecosystems in Bangladesh” describes the prevalence of Escherichia coli and its resistance status in raw vegetables, soil, and water samples collected from rooftop and surface gardens in urban and peri-urban areas of Bangladesh. From different reports, it is evident that family vegetable and fruit gardens on rooftops and surfaces are very common, particularly in city areas in Bangladesh. These gardens are a source of vegetables and a common area for family time. While gathering in gardens, family members, particularly children, sometimes eat vegetables and fruits without washing. However, the subject choice is appreciable as AMR in such a neglected area should be documented for policymakers.

However, a few revisions and corrections will improve its quality. To me, the background of the study requires adding the concept of rooftop and surface gardens in city areas and explaining why city people are growing family gardens on rooftops and surfaces. In addition, please explain what agricultural practices are used in those gardens that could induce AMR in garden components.

Line no. 118: Did the Authors use any sample size calculation formula? How did the Author finalize the 145 number of the sample? Any previous study in Bangladesh?

Line no. 121: In Bangladesh, people generally cook vegetables at high temperatures rather than the salad types of vegetables. Therefore, it is essential to know the names and types of vegetables. Please mention it. In addition, the risk is different for raw and cooked vegetables. What is the source of the water sample? It is better to write a short paragraph about sample types, sources, etc.

Line no. 123, 140: Please make “E. coli” italic and search the whole manuscript and supplementary file for similar errors.

Line no. 131: Please avoid (.) color in writing.

Line no. 140-141: Based on the statement, positive control was used in PCR amplification. Please check the Fig. S2 and revise the figure legend.

Line no. 161: Please mention the incubation period; generally, 16 hours is required to interpret the disc diffusion test.

Line no. 162: Please check the sentence – “multidrug resistance….”

Table 2: To me, in Pearson correlation analysis, when a variable is constant (100% or 0%), that variable is not computed. The prevalence of ampicillin resistance is 100%. Therefore, Pearson correlation for ampicillin is not possible. Please revise Table 2.

Table 3: Table 3 requires colossal revision. As per the statement, 54 isolates were randomly selected for the antimicrobial susceptibility test. Therefore, the number of isolates from different resistance patterns must be 54. However, it is only 17 from Table 3, column 4. Currently, among the 17 isolates, only eight are classified as MRD. Please add all 54 isolates in Table 3. Then, a different finding (maybe the pattern will be more, and the MDR percent will be changed) will be found. In addition, the heading of column 4 is “No. of MDR Isolates (%).” However, the patterns from rows 5 to 10 are not MDR by definition. Furthermore, the percentage in column 4 is incorrect, as they are not computed based on isolate number but antibiotic class. After the revision, there will be massive changes in results and discussion.

Line no. 263-264: This data is not found in the Table 4. Please check for similar errors in the whole manuscript.

Table 4: It is better to find the association of different tet genes in tetracycline resistance similarly to different bla genes for the beta-lactam antibiotics. However, this study does not have that design. Finding an association of tetA in some heterogeneous phenotypic resistance (ampicillin, ciprofloxacin, and others) is not useful and is the same for blaTEM. To establish such a relationship, please point them out in the discussion section with solid references.

Line no. 267 -270: This information is not helpful. Among the eight tetracycline-resistant isolates, seven had blaTEM, which is only 15.9%, based on Table 4, and the Authors are presenting it as a low percentage of blaTEM in tetracycline resistance. However, it could be a maximum of 8 isolates, as the number of tetracycline resistance is 8. Then how could it be 100% or close to 100%? It is not possible. Therefore, this comparison is worthless. I suggest not to keep this table.

Line no. 265: Please make the gene name style uniform.

Please check that the figure legends are missing and the figures are not self-explanatory.

Reviewer #3: 1.The affiliation for co-author number 3 and 4 from Taiwan and Thailand should be completed like others.

2. There were so many factors related to the difference between the rooftop and the surface gardens, thus, how different of soil, water and vegetable preparation among the 3 areas, Since this lack of the elementary information of these 3 areas of gardening for vegetable. Of course, that E. coli could be detected anywhere without any suspicious questions but the results seem not so benefit in term of garden management of vegetable for human consumption.

3. Please added the demographic information of DNCC, DSCC and GCC of how different of soil, water and vegetable growing in these 3 areas. Not just only details of 3 different location which show no meaning for E. coli isolation.

4. The result shown that E. coli from rooftop garden was lower than surface garden, however, reader still do not know how difference of gardening management of these two kinds of garden and pricing of 2 different gardening reported in urban and peri-urban areas.

5. How come only AMR genes of blaTEM and tetA were studied? How about the others? This is the limitation of genotypic study in this research work. Where is malB?

6. This study and the conclusion lacks of management issue after finding the E. coli contamination in Vegetable, of which the contamination normally found in fresh vegetable. Just only mentioned in line between 307-309 is too few.

7. Line 369-371: the recommendation should be more strictly applicable rather than only using potable water to clean the rooftop garden. Any other application for the surface garden should be done as well??

6. Why didn’t author mentioned about malB gene in the genotypic study? It didn’t mention at all so please add details inside.

Reviewer #4: 1. Authors are requested to update the reference list and cite with some recent articles.

2. Line 140. E. coli should be in Italic. Please check this throughout the manuscript.

3. Line 111 Cross-sectional studies of what?? Make it clear

4. What are the microflora of rhizospheric soils? Please identify the soil microflora of each sampling sites and including the findings in the revision.

5. Is there is any horizontal gene transfer between E. coli with plant associated bacteria?

6. Authors are requested to draw a scheme how E. coli adapts to plants defense molecules viz salicylic and jasmonic acid?. How plants induce selection for resistant E. coli?

7. Authors may also check the presence of AvrE, HopZ proteins in any plant system to understand how E. coli-plant interactions.

8. Picture quality and presentation is poor. Please increase the dpi atleast 300 dpi.

9. Why GCC and DSCC rooftop have prevalence of E. coli >70? Explain

6. PLOS authors have the option to publish the peer review history of their article (what does this mean?). If published, this will include your full peer review and any attached files.

Reviewer #1: **Yes: **Professor Sukumar Saha

Reviewer #2: **Yes: **Md. Abdus Sobur

Reviewer #3: No

Reviewer #4: No

---

## [Author Response · Author response to Decision Letter 0]

5 Nov 2024

Dear Editor,

Attached to this submission is our point-by-point responses to the comments raised by both editor and reviewers. We would like to take this opportunity to express our sincere thanks to the expert reviewers/editors who identified several areas in our manuscript that were needed corrections as well as modifications. We also would like to cordially thank you for allowing us the change to resubmit a revised version of the manuscript. 

We have revised and updated the manuscript with some modifications as per reviewers’ suggestion. Please find all changes highlighted in RED color fonts in the revised manuscript. We also have provided a clean manuscript for your kind perusal.

---

## [Editor Report · Decision Letter 1]

4 Dec 2024

Prevalence and antibiotic resistance of Escherichia coli in urban and peri-urban garden ecosystems in Bangladesh

PONE-D-24-32199R1

Dear Dr. Rahman,

We’re pleased to inform you that your manuscript has been judged scientifically suitable for publication and will be formally accepted for publication once it meets all outstanding technical requirements.

Kind regards,

Bilal Aslam, PhD

Academic Editor

PLOS ONE
---

## [Editor Report · Acceptance letter]

19 Dec 2024

PONE-D-24-32199R1 

PLOS ONE

Dear Dr. Rahman, 

I'm pleased to inform you that your manuscript has been deemed suitable for publication in PLOS ONE. Congratulations! Your manuscript is now being handed over to our production team.

Kind regards, 

on behalf of

Dr. Bilal Aslam 

Academic Editor

PLOS ONE